# Evaluating Reinforcement Learning Agents for Anatomical Landmark Detection

**Amir Alansary[1], Ozan Oktay[1,2], Yuanwei Li[1], Loic Le Folgoc[1], Benjamin Hou[1], Ghislain Vaillant[1], Ben Glocker[1], Bernhard Kainz[1] and Daniel Rueckert[1]**

[1]Imperial College London, London, UK
[2]Babylon Health, London, UK
a.alansary14@imperial.ac.uk

## Abstract

Automatic detection of anatomical landmarks is an important step for a wide range of applications in medical image analysis. Manual annotation of such landmarks is a tedious task and prone to observer errors. In this paper, we evaluate novel deep Reinforcement Learning (RL) strategies to train agents that can precisely localize target landmarks in medical scans. An artificial RL agent learns to identify the optimal path to the point of interest by interacting with an environment, in our case 3D images. Furthermore, we investigate the use of fixed- and multi-scale search strategies with hierarchical action steps in a coarse-to-fine manner. Multiple Deep Q-Network (DQN) based architectures are experimented in training of the proposed RL agents achieving good results for detecting multiple landmarks using a challenging fetal head ultrasound dataset.

## 1   Introduction

Accurate detection of anatomical landmarks from medical images is an essential step for many image analysis and acquisition methods. For instance, the localization of the Anterior Commissure (AC) and Posterior Commissure (PC) points in brain images is required to obtain the optimal view of the mid-sagittal plane. This can be used as an initial step for image registration [1] or evaluating pathological brains [2]. Another example, during conventional cardiac MRI acquisition, is the localization of standard views such as 2- and 4-chamber views. This requires a multi-step approach that involves automatic landmark detection [3, 4]. Such view planning is important for consistent evaluation of different patients using standardized biometric measurements [5]. Landmark localization can be also used to initialize deformable models and atlas based approaches for the evaluation of cardiac ventricular functions [6]. Furthermore, in fetal imaging, anatomical landmarks are used for estimating qualitative scores of fetal biometric measurements, such as: fetal growth rate, gestational age, and to identify abnormalities [7]. They are also required in order to identify standardized views such as transventricular and transcerebellar planes, which are commonly used in clinical practice [8].

Since manual landmark annotation is time consuming and error prone, different automatic methods were developed to tackle this task. The design of such methods is challenging due to the variability of the scanned organ's shape, size, orientation, and image quality. Inspired by [9], we formulate the landmark detection problem as a sequential decision process of a goal-oriented agent, navigating in an environment, the medical image, towards a point of interest. At each time step, the agent should decide which direction it has to move to find the target landmark. We use reinforcement learning (RL) to approximate the solution of this sequential decision making process. One of the main advantages of applying RL to the landmark detection problem is the ability to learn simultaneously both searching strategy and the appearance of the object of interest as a unified behavioral task for an artificial agent.

1st Conference on Medical Imaging with Deep Learning (MIDL 2018), Amsterdam, The Netherlands.

It also does not require any hand-crafted features and can be trained end-to-end. RL has the power to perform in a partial field-of-view or incomplete data, which can be useful for real-time applications.

The main contributions of this works can be summarized as follows: (I) We propose and demonstrate use cases of several different Deep Q-Network RL models for anatomical landmark localization in 3D fetal US images. (II) Additionally, we propose a novel fixed- and multi-scale optimal path search strategy with hierarchical action steps for agent based landmark localization frameworks.

## 2 Related work

Typical landmark localization methods can be mainly categorized into three approaches: registration, structure and image based. The first category depends on robust rigid or non-rigid image registration techniques to match corresponding points of interest between target and reference images [10, 11]. Structure based methods rely on spatial priors that capture the location of different landmarks by learning an appearance model [12, 13, 14]. Image based methods learn a set of local image features located around the anatomical landmarks [15].

In the literature, most of the published works have adopted machine learning algorithms for landmark detection by learning a combined appearance and image based model. For example, Criminisi et al. [16] proposed a regression forest based landmark detection framework to locate organs in full-body CT scans, which uses Haar-like appearance features. Despite being fast and robust, this approach achieved less accurate localization results for larger organ structures. Gauriau et al. [17] extended the work of [16] by incorporating statistical shape priors derived from segmentation masks with cascaded regression. Oktay et al. [18] used a stratification based training model for the decision forest, where the latent variables within the stratified trees are probabilistic. Urschler et al. [19] proposed a unified random forest based framework combining appearance information with geometrical distribution of landmark points. These methods achieved robust results for locally similar structures by learning particular hand-crafted features extracted from training data. However, the design of such features requires prior knowledge about the points of interest.

Outwith the success of deep learning in different image-based applications, Zheng et al. [20] exploited a two-stage approach for landmark detection using convolutional neural networks, aptly named ConvNet. The first stage, comprises of a shallow network with one hidden layer, is used to extract a number of 3D point candidates using a sliding window. This is followed by a deeper network, which is applied on image patches extracted around the selected points. However, these methods may fail to localize locally similar structures as it does not account for the global contextual image information. Zhang et al. [21] proposed a similar approach utilizing two ConvNets to learn 3D displacements to a common template, which is followed by another convolutional layer for predicting the coordinates of multiple landmarks jointly. The first network is trained using image patches, whereas the second network shares the same first network architecture and weights as well as adding extra layers. The second network is trained using the whole image instead of patches to learn global information on top of the local information learned by the first network. Payer et al. [22] adopted a ConvNet to model spatial configuration to detect multiple landmarks. The first block of their architecture generates local appearance heatmaps for individual landmark locations. Later, the relative position of a single point with respect to the rest of the landmarks is learned through another convolutional kernel. The final heatmap combines both local appearance and spatial configuration between all landmarks. Although ConvNets based approaches showed superior performance compared to the traditional approaches, increasing the network's field of view requires bigger memory and higher computational complexity. Thus, these approaches may fail to capture spatial relations within a global neighborhood, and are limited to images with lower resolutions.

In order to capture global as well as local information, Andermatt et al. [23] presented a method based on multi-dimensional gated recurrent units combining two recurrent neural networks. The first network detects a candidate region around the point of interest followed by a second network for more accurate localization. Yet this method introduces higher complexity by adding extra parameters to learn for the gated recurrent units. Ghesu et al. [9] adopted a deep reinforcement learning (RL) agent to navigate in a 3D image with fixed step actions for automatic landmark detection. The artificial agent learns the optimized path from any location to the target point by maximizing the accumulated rewards of taking sequential action steps. Xu et al. [24], inspired by [9], proposed a supervised method for action classification using image partitioning. Their model learns to extract an action map

for each pixel of the input image across the whole image into directional classes towards the target point. They use a fully ConvNet with large receptive field to capture rich contextual information from the whole image. Their method achieved better results than using an RL agent, however, it is restricted to 2D or small sized 3D images due to the computational complexity of 3D ConvNets. In order to overcome this additional computational cost, Li et al. [8] presented a 2.5D patch-based iterative ConvNet to detect individual or multiple landmarks simultaneously. Moreover, Ghesu et al. [25, 26] extended their RL based landmark detection to exploit multi-scale image representations.

## 3 Theory

Machine learning enables automatic methods to learn from data and improve from experience to either make a decision or take an action. Broadly, machine learning algorithms can be classified into three main categories: unsupervised, supervised and reinforcement learning. Unsupervised learning methods rely on exploring unlabelled data and drawing inferences to describe hidden structures. Whilst in a supervised manner, the learning is done from a training set of labeled examples provided by a knowledgeable external supervisor. Reinforcement learning (RL) involves learning by interacting with an environment, which allows artificial agents to learn complex tasks that may require several steps to reach a solution [27]. RL has been applied to several medical imaging applications such as landmark detection [9, 25, 26], tissue localization [28] and segmentation [29], image registration [30, 31], and view planning [5]. In this section, we will explain briefly the theory behind RL followed by the application of deep learning to approximate its solution. Then we will explain the proposed deep RL-based framework shown in Fig. 1.

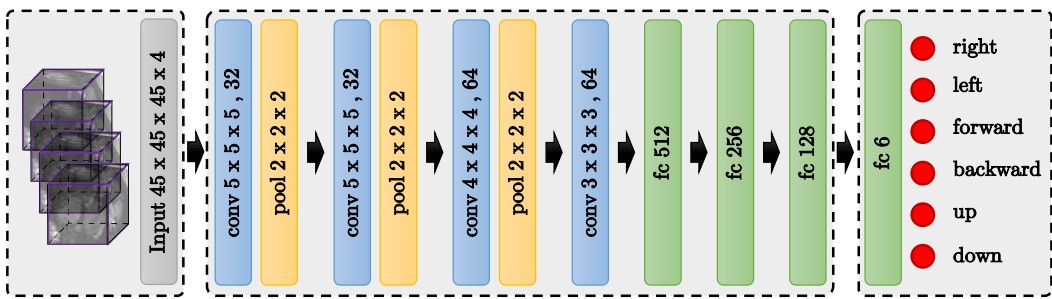

Figure 1: Schematic illustration of the proposed DQN-based network architecture for anatomical landmark detection. The input is the 3D trajectory of the region centered around the point of interest. The output is the approximated $Q$-value for the six possible actions. The agent will pick the action with the highest $Q$-value. This is done sequentially until the agent finds the target landmark.

### 3.1 Reinforcement learning (RL)

Inspired by behavioral psychology, RL can be defined as a computational approach to learn by interacting with an environment so as to maximize cumulative reward signals [27]. A learning agent interacts with an environment $E$; as such at every state $s$, a single decision is made to choose an action $a$ from a set of multiple discrete actions $A$. Each valid action choice results in an associated scalar reward, defining the reward signal, $R$. This sequential decision making can be formulated as a Markov Decision Process (MDP), where each $s_t$ and $a_t$ are conditionally independent of all previous states and actions holding the Markov assumption. The main goal is to learn an optimal policy that maximizes not only the immediate reward but also subsequent future rewards. The optimal function can be computed directly given the whole MDP using dynamic programming. However, in medical imaging problems the MDP is usually incomplete. RL approximates the optimal function iteratively by sampling states and actions from the MDP, and learning from experience. There are several algorithms to solve an RL problem such as certainty equivalence, temporal difference (TD) and $Q$-learning. Because of the recent success of employing Q-learning in several medical imaging applications [5, 9, 25, 26, 28, 29, 30, 31], we adopt in this paper the common strategy of Q-learning based methods as a solution for the RL problem formulation of landmark detection.

### 3.1.1 Q-Learning

Learning an optimal RL policy essentially results in learning to map a given state to an action by maximizing the sum of numerical rewards seen over the agent's lifetime. The optimal action-selection policy can be identified by learning a state-action value function $Q(s, a)$ [32], which measures the quality of taking a certain action $a_t$ in a given state $s_t$. The $Q$-function is defined as the expected value of the accumulated discounted future rewards $E[r_{t+1} + \gamma r_{t+2} + \cdots + \gamma r_{t+n}|s, a]$. $\gamma \in [0, 1]$ is a discount factor that is used to weight future rewards accordingly. It can represent the uncertainty in the agent's environment by providing a probability of living to see the next state. This value function can be unrolled recursively (using the Bellman Equation [33]) and can thus be solved iteratively:

$$Q_{i+1}(s, a) = E\left[r + \gamma \max_{a'} Q_i(s', a')\right]. \tag{1}$$

Where $s'$ and $a'$ are the next state and action. We can find the optimal action for each state by solving the previous equation. The optimal action will have the highest long-term reward $Q^*(s, a)$.

### 3.1.2 Deep Q-Learning

The advent of deep learning has fuelled the currently highly active RL research field. Mnih et al. [34] proposed a deep convolution network to approximate $Q(s, a) \approx Q(s, a; \omega)$, known as deep Q-network (DQN), achieving human-level performance in a suite of Atari games. Approximating the $Q$-value function in this manner allows the network to learn from larger data sets using mini-batches. A naive implementation of DQN will suffer from instability and divergence issues because of: *(i)* the correlation between sequential samples, *(ii)* rapid changes in $Q$-values and the distribution of the data, and *(iii)* unknown reward and $Q$-values range that may cause large and unstable gradients during backpropagation. In [34], they proposed to use a target $Q(\omega^-)$ network that is periodically updated with the current $Q(\omega)$ every $n$ iterations. Freezing the target network during training stabilizes the rapid policy changes. To avoid problem of successive data sampling, an experience replay memory $(D)$ [35] can be used to store transitions of $[s, a, r, s']$ and randomly sampling mini-batches for training. The approximation of best parameters $\omega^*$ can be learned end-to-end using stochastic gradient descent (SGD) of the derivative of the DQN loss function $\frac{\delta L(\omega)}{\delta \omega}$, where:

$$L_{DQN}(\omega) = E_{s,r,a,s' \sim D}\left[\left(r + \gamma \max_{a'} Q(s', a'; \omega^-) - Q(s, a; \omega)\right)^2\right]. \tag{2}$$

In order to prevent $Q$-values from becoming too large, also to ensure that gradients are well-conditioned, rewards $r$ are clipped between $[-1, +1]$. This trick works for most of the applications in practice, however, it may have the drawback of not differentiating between small and large rewards. We outline below two recent state-of-the-art improvements to the standard DQN, and evaluate them experimentally in Section 4.

### 3.1.3 Double DQN (DDQN)

In noisy stochastic environments, DQN [34] sometimes significantly overestimates the values of actions [36]. This is caused by a bias introduced from using the maximum action value as an approximation for the maximum expected value. The max operator, $max\ Q(s', a'; \omega^-)$, uses the same values to select and evaluate an action resulting in selecting overestimated (overoptimistic) values. Van Hasselt et al. [36, 37] proposed a solution, double DQN (DDQN), to mitigate bias by decoupling the selected action from the target network. Thus current network is used for the action selection resulting in a modified loss function:

$$L_{DDQN}(\omega) = E_{s,r,a,s' \sim D}\left[\left(r + \gamma \max_{a'} Q(s', Q(s', a; \omega); \omega^-) - Q(s, a; \omega)\right)^2\right]. \tag{3}$$

DDQN improves the stability of learning and may translate to the ability to learn more complicated tasks. The results of DDQN [37] illustrate reduction in the observed overestimation and better performance than DQN [34] on several Atari games.

### 3.1.4 Duel DQN

Q-values corresponds to the quality of taking a certain action given a certain state $Q(s, a)$. Wang et al. [38] proposed to decompose this action-state value function into two more fundamental notions of value. The first is an action-independent value function $V(s)$ to provide an estimate of the value of each state without having to learn the effect of each action. The second is an action-dependent advantage function $A(s, a)$ to calculate potential benefits of each action. Intuitively, the $Q$-function learns separately how good is a certain state and how much better taking a certain action would be compared to the others. The new combined dueling DQN function is defined as:

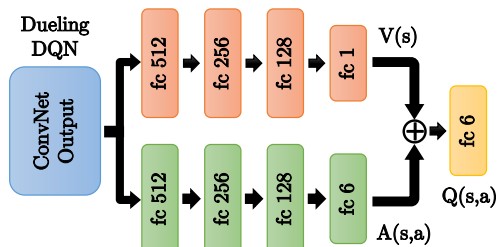

Figure 2: Duel DQN architecture, which splits the fully connected (FC) layers into two paths: the state value $V(s)$ and action advantage $A(s, a)$ functions.

$$Q(s, a) = A(s, a) + V(s). \qquad (4)$$

This is can be implemented by splitting the fully connected layers in the DQN architecture to compute the advantage and state value functions separately, then combining them back into a single Q-function only at the final layer with no extra supervision, see Fig. 2. Duel DQN can achieve more robust estimates of state value by decoupling it from specific actions, $s$ is more explicitly modelled, which yields higher performance in general. Duel DQN [38] showed better results than the previous baselines of DQN [34] and DDQN [37] on several Atari games. Broadly, duel DQN and DDQN introduced vast improvements in performance compared to DQN, yet it does not necessarily result in better performance in all environments.

## 3.2 RL agents for landmark detection

In this work, inspired by [9], we formulate the problem of landmark detection as an MDP, where an artificial agent is learned to make a sequence of decisions towards the target point of interest. In this setup, the input image defines the environment $E$, in which the agent navigates using a set of actions. The main goal of the agent is to find an anatomical landmark. In this section, we explain the main elements of the MDP that includes set of actions $A$, set of states $S$, and reward function $R$. During testing, the agent does not receive any rewards and does not update the model either, it just follows the learned policy.

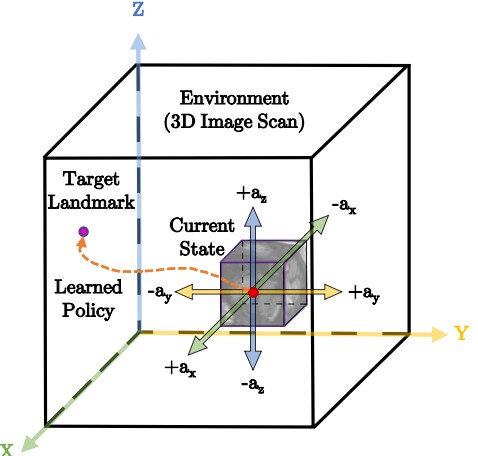

Figure 3: Schematic diagram of the proposed RL agent interacting with the 3D image environment $E$. At each step the agent takes an action towards the point of interest. These sequential actions forms a learned policy forming a path between the starting point and the target landmark.

### 3.2.1 Navigation actions

The agent interacts with $E$ by taking movement action steps $a \in A$ that imply a change in the current point of interest location. The set of actions $A$ is composed of six actions, $\{\pm a_x, \pm a_y, \pm a_z\}$, in the positive or negative direction of $x$, $y$ or $z$. For instance, taking a $+a_x$ action means that the agent will

move a fixed step size in the positive $x$-direction. Figure 3 shows a schematic visualization of these navigation actions in a 3D scan.

### 3.2.2 States

Our Environment $E$ is represented by a 3D image, where each state $s$ defines a 3D region of interest (ROI) centered around the target landmark. A frame history buffer is used to capture the last 4 action steps (ROIs) taken by the agent in its search for the landmark. This stabilizes the search trajectories and prevent the agent from getting stuck in repeated cycles.

### 3.2.3 Reward function

Designing good empirical reward functions $R$ is often difficult as RL agents can easily overfit the specified reward and thereby produce undesirable or unexpected results. For our problem, the difficulty arises from designing a reward that encourages the agent to move towards the target plane while still being learnable. Thus, $R$ should be proportional to the improvement that the agent makes to detect a landmark after selecting a particular action. Here, similar to [9], we define the reward function $R = D(P_{i-1}, P_t) - D(P_i, P_t)$, where $D$ represents the Euclidean distance between two points. We further denote $P_i$ as the current predicted landmark's position at step $i$, with $P_t$ the target ground truth landmark's location. The difference between the two Euclidean distances, the previous step and current step, signifies whether the agent is moving closer to or further away from the desired target location. The case $R = 0$ presents oscillations around the correct solution.

### 3.2.4 Terminal state

The final state is reached when there are no further transition states for the agent to take. This means that the agent has found the target landmark $P_t$. We define the terminal state during training when the distance between the current point of interest and the target landmark are less than or equal to 1mm. Finding a terminal state during testing is more challenging, due to the absence of the landmark's true location. One solution is to define a new trigger action that terminates the sequence of the current search when the target state is reached [28, 39]. Although this modifies the environment by marking the region that is centered around the correct location of the target landmark, it increases the complexity of the task to be learned by increasing the action space size. It also introduces a new parameter, maximum number of interactions, which needs to be set manually. It may also slow down the testing time in cases where the terminal action is not triggered. In this work, we adopt the oscillation property to terminate the search process during testing, however, in contrast to [9], we choose the terminating state based on the corresponding lower $Q$-value. We find that $Q$-values are lower when the agent is closer to the target point and higher when it is far. Intuitively, the DQN encourages awarding higher $Q$-values to actions when the current state is far away from the target landmark.

### 3.2.5 Multi-scale agent

DQN-based approaches rely on deep ConvNets to achieve superior results, however, increasing the network's field of view requires larger memory and higher computational complexity. Thus, it will be difficult to capture spatial relations within a global neighborhood. In this work, we adopt a multi-scale agent strategy [25, 26] in a coarse-to-fine fashion with hierarchical action steps. The environment $E$ samples a fixed size image-grid with initial spacing $(S_x, S_y, S_z)$ mm around the current point of interest $P_o$, and the agent searches for the target landmark with initial higher action steps. Once the target point is found, $E$ samples the new image-grid with smaller spacing, as well as the agent uses smaller action steps. Coarser levels in the hierarchy provide additional guidance to the optimization process by enabling the agent to see more structural information. Finer scales, on the other hand, provide more precise adjustments for the final estimation of the plane. Similarly, larger action steps speed convergence towards the target plane, while smaller steps fine tune the final estimation of plane parameters. The same DQN is shared between all levels in the hierarchy.

# 4 Experiments and results

In this work, the performance of different RL agents for anatomical landmark detection is evaluated on fetal head ultrasound images. Finding the target landmarks in such images is a challenging task because of ultrasound artifacts such as shadowing, mirror images, refraction, and fetal motion. We also evaluate fixed- and multi-scale search strategies by sampling with different spacing values. We use 3-levels for the multi-scale agent with spacing values from 3mm to 1mm, decreasing by one at each level. Hierarchical action steps are start from 9 to 1 steps per iteration, dividing by 3 at each level of the hierarchy. We fix the initial selected points for all models during testing for a fair comparison between different variants of the proposed method. In order to report more robust results, we select 19 different starting points distributed in the whole image for every testing subject. We measure the accuracy based on the distance error between detected and target landmarks. Finally, we run extensive comparison between different DQN-based architecture, namely DQN, DDQN, Duel DQN, and Duel DDQN.

**Dataset:** 72 fetal head ultrasound scans [8], randomly divided into 21 and 51 images for training and testing, respectively. We choose three landmarks, right and left cerebellum and cavum septum pellucidum, that are used to define the transcerebellar (TC) plane commonly used for fetal sonographic examination, see Fig. 4. The selected landmarks were manually annotated by clinical experts using three orthogonal views. All images were roughly aligned to the same orientation and re-sampled to isotropic 1mm spacing.

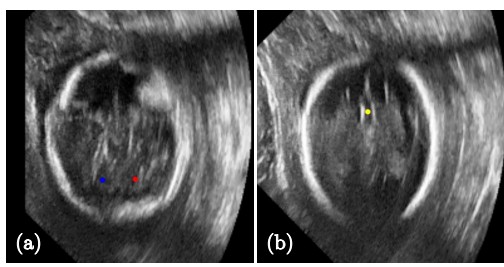

Figure 4: Sample 2D images from fetal head ultrasound showing the target landmarks: (a) right (red) and left (blue) cerebellum, and (b) cavum septum pellucidum (yellow) points.

**Experiments:** During training, a random point is sampled from a region with size $80\%$ of the whole image dimensions around the center. An ROI of size $45x45x45$ voxels is sampled around the selected point. The agent follows an $\epsilon$-greedy exploration strategy, where at every step it selects an action uniformly at random with probability $(1 - \epsilon)$. Every trial to find the point of interest is called an *episode*. Here we use 1500 frames to limit the maximum number of frames per episode. During testing, the agent follows the learned policy by selecting the action with highest $Q$-value at each step.

Table 1: Comparison of different DQN-based agents using fixed-scale (FS) and multi-scale (MS) search strategies for the detection of right and left cerebellum, and cavum septum pellucidum landmarks in fetal ultrasound images. Distance errors are in $mm$.

| Model | Right Cerebellum | | Left Cerebellum | | Cavum Septum Pellucidum | |
|---|---|---|---|---|---|---|
| | **FS** | **MS** | **FS** | **MS** | **FS** | **MS** |
| **DQN [9, 25, 26]** | $4.17 \pm 2.32$ | $3.37 \pm 1.54$ | $2.78 \pm 2.01$ | $3.25 \pm 1.59$ | $\mathbf{4.95 \pm 3.09}$ | $\mathbf{3.66 \pm 2.11}$ |
| **DDQN** | $3.44 \pm 2.31$ | $3.41 \pm 1.54$ | $2.85 \pm 1.52$ | $2.95 \pm 1.00$ | $5.01 \pm 2.84$ | $4.02 \pm 2.20$ |
| **Duel DQN** | $\mathbf{2.37 \pm 0.86}$ | $3.57 \pm 2.23$ | $\mathbf{2.73 \pm 1.38}$ | $\mathbf{2.79 \pm 1.24}$ | $6.29 \pm 3.95$ | $4.17 \pm 2.62$ |
| **Duel DDQN** | $3.85 \pm 2.78$ | $\mathbf{3.05 \pm 1.51}$ | $3.27 \pm 1.89$ | $3.50 \pm 1.7$ | $5.12 \pm 3.15$ | $4.02 \pm 1.55$ |

**Results:** Table 1 shows the comparative results of the performance of different agents. In general, all methods share similar performance including speed and accuracy. However, Duel DQN achieves the best accuracy detecting the right and left cerebellum points, whilst DQN performs the best for finding the cavum septum pellucidum point. Additionally, the multiscale strategy improves the performance of the agent and reach faster to the target point thanks to the hierarchical action steps. More visualizations are on our github repository (https://github.com/amiralansary) showing different real-time examples (videos) of trained agents searching for the target landmarks.

## 4.1 Implementation

Training times are around $24 - 48$ hours for individual landmarks using an NVIDIA GTX 1080Ti GPU. In our implementation we use batch size of $48$, experience replay memory of size $1e5$, activation function PReLU for convolutional layers and leakyReLU for fully connected layers, adam optimizer, $\gamma = 0.9$, and $\epsilon = 0.9 - 0.1$. Figure 1 shows the architecture of the proposed DQN. The source code of our implementation is publicly available on github at https://github.com/amiralansary.

## 5 Conclusion and discussion

In this paper, we proposed different reinforcement learning agents based on DQN architectures for automatic landmark detection. These RL-agents were capable to find automatically the point of interest in real-time by moving towards the target sequentially step-by-step. The performance of the the proposed method was evaluated on challenging fetal head ultrasound images. We also exploited fixed- and multi-scale optimal path search strategies. Despite the fact that RL is a difficult problem that needs a careful formulation of its elements such as states, rewards and actions, our results demonstrate high detection accuracy in noisy ultrasound images. Yet finding the optimal DQN architecture for achieving the best performance is environment-dependant.

**Future work:** We will investigate using intrinsic geometry instead of intensity patterns for the RL environment to improve the performance. Multi-landmark detection is another interesting application to be explored using either multiple competitive or collaborative agents. Another future direction will be to investigate involving human experts for learning the artificial agents actively, inspired by AlphaGo [40], where the agents can learn from experienced operators by interaction and accumulate this experience.

### Acknowledgments

We thank the volunteers, radiographers and experts for providing manually annotated datasets and NVIDIA for their GPU donations. Amir Alansary is awarded the Imperial PhD president's scholarship.

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
