# OpenReview forum: "Evaluating Reinforcement Learning Agents for Anatomical Landmark Detection"
_MIDL.amsterdam/2018/Conference — MIDL 2018 Oral_

### Review · AnonReviewer2 · 2018-05-04
**Good paper**

**Rating:** 4
**Confidence:** 2

**Review:**

Pros:
+clear presentation
+good related work review
Cons:
-the motivation on why to use RL-based methods for landmark detection could be included

The paper evaluates DQN and its variants for the task of landmark localization in head ultrasounds. The paper is well presented and easy to follow. The related work section is well written. However, please note that I'm not an expert in RL.

The one thing I'm missing in the introduction section are comments on why RL-based approaches would be the "best" way of approaching landmark detection in medical imaging.

For GitHub link, it would be better to use the full link to the project: https://github.com/amiralansary/tensorpack-medical/tree/master/examples/LandmarkDetection/DQN

There is a typo on page 4: "There is can be..."

It is important to note that this is not the first paper applying RL to landmark localization in the context of medical imaging (see [9]). Thus, the originality of the work is rather incremental. However, I think that this work might be of interest for Medical Imaging community and I would recommend the paper for acceptance.

**Special Issue:**

Yes

---

### Review · AnonReviewer1 · 2018-05-09
**A very good paper**

**Rating:** 4
**Confidence:** 3

**Review:**

Overall:
The paper proposes to use deep reinforcement learning for automatic anatomical landmark detection. The authors applies it to 3D fetal head ultrasound scans. In general, the paper proposes a new and novel approach to the problem that opens new potential research directions.

Strengths:
+ The paper is written in an easy-to-read manner. All concepts are clearly explained.
+ Typically, an anatomical landmark detection is accomplished through two steps. The paper proposes a new direction to train the procedure end-to-end by utilizing reinforcement learning. It is extremely interesting and novel approach.
+ The presented approach is sound.
+ The results are solid.

Remarks:
* Minor
- It would be beneficial to see what are wall-clock times during testing in order to properly assess whether the proposed approach could work in real-time.

**Special Issue:**

Yes

---

### Review · AnonReviewer3 · 2018-05-12
**Interesting approach to landmark detection**

**Rating:** 4
**Confidence:** 2

**Review:**

The paper proposes to use RL to detect anatomical landmarks and apply it to 3D fetal head images. I enjoyed reading it.

Pros:
* Well written
* Proper evaluation
* Applying RL to medical images is reasonably novel, opens opportunities for new research directions.
* A reasonable novel approach

Cons:
* Why would RL perform on-par or better either in clock times or in detection accuracy compared to other state of the art methods such as e.g. regional CNNs where landmark detection is phrased as a keypoint detection problem?
* I am not an expert in fetal ultrasound but there is mentioned the images get resampled to 1mm - how large are the images? In the first part of the paper the authors write CNN approaches might not capture enough global context, but how much global context is actually needed? There are many ways such as dilated and separable convolutions nowadays to increase the receptive field while keeping the number of parameters, and thus memory, low. The results of the paper do not convince me that this is the best approach.
* Why is deep Q-learning selected? There are several other RL methods such as LA3C.


**Special Issue:**

Yes

---

### Comment · ~Bram_van_Ginneken1 · 2018-05-18
**Selection for longlist for special issue Medical Image Analysis**

Dear authors,

Congratulations on your acceptance to MIDL! We have selected your paper on the longlist for the Medical Image Analysis Special Issue. Please read this page:
https://midl.amsterdam/special-issue-in-medical-image-analysis/
Please answer the three questions that are listed on that page about your interest in submitting to the special issue, potential overlap with other publications, and related publications.

You can post your answer here directly below on openreview.net, or mail me directly at bram.vanginneken@radboudumc.nl.

Best regards, Bram

---

> ### Comment · ~Amir_Alansary1 · 2018-05-30
> **Response to selection for longlist for special issue Medical Image Analysis**
>
> Thank you for selecting our paper for the special issue. Below our answers for the asked questions:
>     1- Yes, we are willing to augment the content of our paper significantly
>     2- Our paper, or any other paper that overlaps with, is and will not be under review or consideration elsewhere
>     3- We are happy to send any related publications from our research group to the reviewers

---

### Decision · Program_Chairs · 2018-05-15
**Paper108 Acceptance Decision**

Oral